# The Effects of Adding a Gel-Alike *Curcuma longa* L. Suspension as Color Agent on Some Quality and Sensory Properties of Yogurt

**DOI:** 10.3390/molecules27030946

**Published:** 2022-01-30

**Authors:** Angélica Serpa Guerra, Catalina Gómez Hoyos, Jorge Velásquez-Cock, Lina Vélez, Piedad Gañán, Robin Zuluaga

**Affiliations:** 1Facultad de Ingeniería Agroindustrial, Universidad Pontificia Bolivariana, Medellin 05004, Colombia; angelica.serpa@upb.edu.co (A.S.G.); lina.velez@upb.edu.co (L.V.); 2Programa de Ingeniería en Nanotecnología, Universidad Pontificia Bolivariana, Medellin 05004, Colombia; catalina.gomezh@upb.edu.co (C.G.H.); jorgeandres.velasquez@upb.edu.co (J.V.-C.); 3Facultad de Ingeniería Química, Universidad Pontificia Bolivariana, Medellin 05004, Colombia; piedad.ganan@upb.edu.co

**Keywords:** *Curcuma longa* suspension, color agent, cellulose nanofibers, yogurt

## Abstract

Color is an important characteristic of food products. This characteristic is related to consumer acceptability. To use the entire rhizome of *Curcuma longa* (CL) as a food colorant, a novel gel alike stable suspension (CLS) was previously developed using cellulose nanofibers (CNFs). Therefore, the present study was conducted to evaluate the CLS as a color additive on a stirred yogurt. Three concentrations of CLS were studied (0.1, 0.125, and 0.15 wt. %) and compared to yogurt without CLS. The obtained yogurts were characterized through the determination of pH, titratable acidity, syneresis, color and curcumin content after 1, 7, 14, and 21 days of storage. Additionally, rheological and sensory measurements were performed on the samples after one day of storage. Results show that the addition of CLS does not affect the pH and titratable acidity of the samples, but all the yogurts showed an increase in their syneresis during the storage time, showing a breakdown of the gel structure. Furthermore, the CLS suspension has the ability to impart a yellow color to yogurts, a characteristic that was stable during storage. Finally, the addition of 1 wt. % or 1.25 wt. % of CLS allows the development of a yogurt with adequate sensory perception.

## 1. Introduction

Turmeric (*Curcuma longa* L.) (CL) is a perennial plant of the ginger family (Zingiberaceae) [1]. Its rhizomes are dried and grounded to obtain turmeric powder, a yellow and orange product that is used to extract curcumin, the phenolic compound responsible for the biological properties associated with turmeric [2]. Additionally, curcumin is a natural color agent known as E 100 [3] that has been used to replace Yellow 5 or tartrazine, a synthetic color agent that has been banned in Norway and Austria [4] while in the European Union, additional labeling information is required for its application in food [5]. Despite the potential of curcumin as a color additive for the food industry, its application is limited because of its low aqueous solubility [6] and the high waste generation associated with its extraction, taking into account that curcumin only represents 3.5 wt. % of the dried turmeric rhizome.

According to the aforementioned information, new alternatives that allow the application of the raw rhizome as a color agent are important to increase the use of turmeric in the food industry. Previous investigations report the development of CL suspensions using ultra-fine friction grinding (UFFG) [7] and the application of cellulose nanofibers (CNF) to stabilize the vegetable cell tissues present in the insoluble phase, obtaining a soft gel alike CL suspension (CLS) [8] that could be used as a natural color agent.

However, the incorporation of new natural color agents in food has different challenges, including its stability during storage conditions and the presence of undesirable odor and flavor in the final product [9], since consumers are looking for natural colorants but expect their products to have additives that relate to the flavors described in their packages [10].

Accordingly, the present study aimed at evaluating CLS as a color additive in a stirred yogurt. For the first time, the soft gel suspension was added to a food product, for these three concentrations of the CLS suspension were evaluated. pH, titratable acidity, syneresis, color, and curcumin concentration were evaluated after 1, 7, 14, and 21 days of storage. Additionally, rheological and sensory measurements were performed on the samples after one day of storage.

## 2. Materials

The CLS suspension was prepared according to the method that was reported [8]. Briefly, 30 wt. % of CL and 0.9 wt. % CNFs were added to water. The mixture was homogenized at 3000 rpm with Ultrafine Friction Grinding (Supermasscolloider MKCA6-2, Mazuco Sangyo, Kawaguchi, Japan) using a Gap or distance between the grinding stones of −1.5. For the yogurt formulations, a natural base yogurt (milk, dried milk, *Streptococcus thermophilus* and *Lactobacillus bulgaricus*) [11,12] was purchased at a local supermarket in Medellin, Colombia was used. 

### 2.1. Yogurt Preparation

Three concentrations of the CLS were evaluated (1 wt. % CLS, 1.25 wt. % CLS and 1.5 wt. % CLS) according to previous experiments carried out with commercial yellow/orange yogurts available in Colombian dairy products market. Results were compared to a yogurt without CLS (0 wt. %). The incorporation of the CLS in the yogurt was performed by adding the suspension and stirring for 3 min (RW-20, IKA, Staufen, Germany) at a low speed to avoid the introduction of air [13]. Batches of yogurt were placed on individual 1000 mL jars. Additionally, for the syneresis determination, 25 g of each yogurt was placed in 50 mL centrifuge tubes. All the samples were stored under refrigeration conditions (4 °C).

### 2.2. Characterization of the Yogurts

Characterization of the samples was performed immediately after production (pH, titratable acidity and color) and after 1, 7, 14 and 21 days of storage (pH, titratable acidity, color, syneresis and curcumin content). Additionally, rheological properties and sensory characterization were performed to the suspensions after one day of storage. 

### 2.3. Ph and Titratable Acidity

Yogurts pH was determined with a pH-meter while the titratable acidity was expressed as g of lactic acid per 100 g of yogurt [14], after titration with 0.1 N sodium hydroxide solution until pH 8.3 ± 0.01. 

### 2.4. Syneresis

Syneresis of the yogurts was reported as the grams of the whey separated out of the total weight of the yogurt. For determination, samples were centrifuged at 3500× *g*. The whey separated from yogurt sample was decanted off and weighed. Syneresis was calculated using Equation (1) [15].
(1)$$\;\mathrm{WHC}=\frac{g\;\mathrm{of}\;\mathrm{whey}}{g\;\mathrm{of}\;\mathrm{yogurt}}*100\%\;$$

### 2.5. Color

Color was measured at CIEL*a*b coordinates using a colorimeter (SP60, X-rite, Grand Rapids, MI, USA). On the CIEL*a*b system L* corresponds to brightness, a* is the chroma of green to red and b* is the chroma of blue to yellow [16]. Additionally, the total color difference (DE) was calculated using Equation (2), where L_0_*, a_0_*, and b_0_* correspond to the coordinates of the samples at day 0 of storage.
(2)$$\Delta E=\sqrt{{\left(L^{*}-L_{0}{}^{*}\right)}^{2}+{\left(a^{*}-a_{0}{}^{*}\right)}^{2}+{\left(b^{*}-b_{0}{}^{*}\right)}^{2}}$$

### 2.6. Curcumin Quantification

Determination of curcumin content of the yogurts was achieved using Ultra-High Pressure Liquid Chromatography [17]. For the extraction 1 g of the freeze-dried yogurt was mixed with 100 mL of methanol and submitted to sonication at room temperature. The diluted supernatant was filtered with a 0.25 mm disc filter and injected into an ultra-high pressure chromatograph (Ultimate 3000 UHPLC, Thermo Fisher Scientific, Waltham, MA, USA) coupled to a UV-Vis detector (Ultimate 3000 VWD, Thermo Fisher Scientific, Waltham, MA, USA). The movil phase (0.6 mL/min) was acetonitrile/phosphoric acid (55:45). The components were separated using an Accucore polar premium column (length 100 mm, internal diameter 3 mm; 2.6 µm particle size) (Thermo Fisher Scientific, Waltham, MA, USA), and the measurement was performed at 428 nm, using the calibration curve (R^2^ = 0.998) plotted for a standard solution of curcumin (Sigma-Aldrich, Saint Louis, MO, USA) at different concentrations. 

### 2.7. Rheology

The rheological characterization of the samples was performed by triplicate using a HR-2 rheometer (T.A. Instruments) using the plate-plate geometry at 4 °C [18]. Data acquisition and processing were performed with the TRIOS software (Version, TA Instruments Ltd., New Castle, DE, USA). To determinate flow behavior, the shear stress was recorded at increasing shear rates (upward flow curve) followed by decreasing shear rates (downward flow curve) [18] after submitting the samples to shear rate ranging from 0.1 to 100 s^−1^. The upward curve was fitted using the Ostwald-de-Waele model (Ec. 3), where η is the apparent viscosity (Pa.s); $\dot{\gamma }$ is the shear rate (s^−1^), K is the consistency index (Pa.s^n^) and n is the flow index (dimensionless). Additionally, hysteresis loop area ΔA, and apparent viscosity $\eta_{\mathrm{app}}$ at a shear rate of 50 s^−1^ were obtained [18].
(3)$$\eta =K\;{\dot{\gamma }}^{n-1}$$

### 2.8. Sensory Evaluation

To evaluate the sensory properties of the obtained yogurts, an acceptance sensory analysis was performed with a 9-point hedonic scale, being 1: dislike extremely and 9: likely extremely [19,20,21]. A total of 40 untrained panelists with ages ranging from 19 to 33 were recruited from the Universidad Pontificia Bolivariana in Medellin, Colombia. For the quantification of the descriptive attributes of the samples, an equal amount of sucrose (7 wt. %) and mango flavor was added before the sensory evaluation to reduce the bitterness of the rhizome. The evaluated attributes were color, flavor, texture, aroma, and overall acceptability. The samples were given to the panelist in plastic cups (20 mL) coded with a three digits code. The panelist used water and unsalted cookies to clean their palates between samples [22]. Given that the evaluated yogurt samples included a nanosized component (CNFs), the participation was voluntary, and an explication of its implications in terms of normative and security was performed to each panelist before the assay. To continue, each one had to sign an informed consent. 

### 2.9. Statistical Analysis

Experimental tests were performed in triplicate. Results are given as the mean of the 3 measurements and the standard deviation (±). Statistical analysis was performed using one-way analysis of variance (ANOVA) using Statgraphics Centurion 18 (Statgraphics Technologies Inc., The Plains, VA, USA). Significance was determined at 95% of confidence. A *p*-value < 0.05 was considered significant. 

## 3. Results and Discussion

### 3.1. Ph and Titratable Acidity

Yogurt is a dairy product that is obtained after the fermentation of milk through starter cultures. Thus, the determination of pH and titratable acidity are parameters used to define its quality [23]. With acidification, the milk casein coagulates and precipitates, generating the characteristic texture of this product. Additionally, the lactic acid in yogurts acts as a preservative against some unwanted bacteria in the final formulation [24]. Given the importance of these two properties during yogurt processing, the results of pH and titratable acidity of the obtained yogurts are presented in Table 1. 

The measure of pH and titratable acidity are complementary properties in yogurts development since pH falls in response to the conversion of lactose to lactic acid. The pH defines the endpoint of the fermentation and must be between 4 and 4.6 [25]. According to the obtained results, all the samples’ pH was in this range. Thus, they have the appropriate pH for this type of product. Additionally, storage conditions did not generate a statistically significant change in this property. That is, during storage under refrigeration conditions, no significant conversion of the lactose in lactic acid was achieved. This behavior is confirmed by the results of titratable acidity of the samples, where the observed increase is not statistically significant. If yogurts are added with fruits or fiber, the pH and titratable acidity must be monitored to establish possible variations induced by the new ingredient [26]. In this particular case, the CLS has a neutral pH value (7.07 ± 0.23). According to the results, the addition of CLS in yogurts did not generate significant changes in pH and acidity. This behavior could be related to the amount of CLS added to the yogurts, since it was very low compared to other studies where changes of the pH were reported after adding new ingredients to the yogurts [22,26]. The observed behavior is important since changes in these properties are not desired during the storage of fermented milk, as they can lead to changes in the physical and sensory properties of the developed product.

### 3.2. Syneresis

Syneresis is a physical property considered a defect in yogurts [27]. It represents the expulsion of the aqueous phase that includes water trapped in the structures of the product, water linked to proteins, and free water [28]. Syneresis is the response of the rearrangement capacity of the bonds in the network and the balance between the repulsion and attraction forces in the casein network [29]. The results of the syneresis of the yogurts during storage time is presented in Figure 1. 

According to the obtained values, all the samples showed an increment in their syneresis during storage time. However, for the sample without CLS, the results were not statistically representative (*p* < 0.05), which allows establishing that the addition of CLS may generate the breakdown of the gel structure of the yogurt [30]. This behavior is more representative in the last days of storage, where the highest values are reached, and all the samples showed the same syneresis value (day 21). This behavior can be associated with structural differences between the yogurt and the CLS. In the yogurt, the balance of the gel is achieved by the attraction and repulsion forces established during the casein precipitation in the fermentation process [29], while in the CLS suspension, the stabilization and gel formation is achieved by the three-dimensional network of cellulose nanofibers that were added during its development [31]. Thus, the addition of the suspension in the yogurt generates an imbalance in the water-protein ratios formed during fermentation [32], given the water present in the CLS. Additionally, it is important to establish that the suspension has different compounds, including water and starch [8,33], which can contribute to an increase in the rearrangement of the bonds in the casein network [27], that will result in larger pore sizes in the yogurt as storage continues, generating greater serum releases in the final days of storage [34]. This behavior has been reported in yogurts that are added with different ingredients, including grapes and coffee [27,34], where higher content of the extract was associated with an increase in the syneresis during storage. Furthermore, during the development of fruit-added yogurts, an increase in its syneresis is evidenced in response to the breakdown of the protein gel by the rearrangement of the casein network [32]. Since this behavior is undesirable in yogurts, the addition of gums such as xanthan and carrageenan seem to lower syneresis during storage [35]. The observed effect when using carrageenan is associated with the interaction between its charges and the ones on the surface of casein micelles that strengthen the casein network [36]. On the other hand, for neutral hydrocolloids like xantan, the reduction of syneresis is associated with an increase in the viscosity of the continuous phase. As mentioned before, the CLS was developed using CNFs as a stabilizer. Their use in the food industry has increased, given its rheological behavior derived from their possibility to develop an entangled three-dimensional network and its characteristic aspect ratio [37]. However, as established in the literature, to achieve the formation of this network, a minimum percentage of CNFs is required [38]. In the particular case of this investigation, the final CNFs content in the yogurts is very low (0.009–0.015 wt. %) compared to values reported in the literature for this type of product, between 0.3 and 1 wt. % [39,40], which allows saying that the effect of this nanocomponent is not representative in the stability behavior of the final product, and the observed one is associated with all the components in the CLS suspension, including water and starch, that interact with the casein network contributing to the separation of the serum.

### 3.3. Color

From a qualitative point of view, Figure 2 shows that all the formulations have similar colors, and according to the image, the storage of the samples did not affect this property.

However, since human eyes are subjective and different physiological factors may influence color perception [41], the samples were characterized by determining their CIEL*a*b* values. Table 2 shows the parameters L* (lightness), a* (redness), b* (yellowness), C (chroma) and ΔE (total color change) of the suspensions.

Results show that the 0% CLS sample presents higher values of brightness and a tendency for green colorations (negative values of a*) related to the oxidation of the fatty acids and the proteolytic activity characteristic of yogurts [42]. Furthermore, this sample has yellow colorations (positive values of b*) related to the milk carotenoids [32]. Additionally, for this sample, the storage only affected the yellow color as a response to the degradation of the β-carotene in milk [30]. All the color coordinates obtained for the yogurt without CLS agree with the ones reported in the literature for this product [43]. As for yogurts added with CLS, they presented high values of brightness while its green colorations tend to reduce when the CLS concentration increase. This behavior is associated with the color of the CLS since this suspension has red-yellow colorations [8] that reduces the tendency for green coloration in the yogurt. Additionally, the green coloration of the yogurts can also be related to fatty acid oxidation [42] and to the presence of riboflavin in the milk (Walstra et al., 2006). The b* coordinate of color in these samples increased with the concentration on CLS, indicating stronger tendencies for yellow colorations. This behavior shows the potential of the suspension as a color additive. Additionally, the results obtained during the storage time showed that the color of the samples, in general, was stable. Thus, the CL suspension was stable in this type of product. The observed stability can be associated with the protective effect of fat and proteins in the yogurt; although the mechanism of this effect is not clear, it could be related to the interaction with the hydroxyl groups [44]. These color results have shown that the use of the entire turmeric rhizome as a color additive is possible through the development of a suspension using cellulose nanofibers since previews reports that evaluate turmeric powder as a color additive in yogurts have shown that the obtained color was not homogeneous and could de unstable through storage time [11]. The stability of the color during storage time of the yogurts added with CLS can be seen in the total color change values (ΔE) of the yogurts, Figure 3. 

The total color change (ΔE) results show that the main changes during storage time are in the first seven days; after this day, the changes are not statistically significant. Additionally, the changes of color associated with the concentration of CLS were not significant either. However, despite the obtained changes during the first days of storage, the values have magnitudes lower than 4, which means that those color changes are not perceptible by an average observer [45], as mentioned before during the qualitative assessment of color. As stated before, curcumin is the compound responsible for the yellow color of Curcuma longa products, its quantification in the samples was performed. 

### 3.4. Curcumin Content

The amount of curcumin in the yogurts with and without CLS are presented in Figure 4.

According to the results of the curcumin content in the yogurts, the concentration of this compound increased with the concentration on CLS as a response to a higher concentration of CL. Furthermore, the results during the storage time allow establishing that the degradation of curcumin is not significant. This behavior is related to the protective effect of the proteins and fat in milk with curcumin [46]. As mentioned before, this effect could be related to the interaction with the hydroxyl groups [44]. Additionally, given the importance of curcumin as an exogen antioxidant, it is important to establish that the amount of curcumin in the yogurts is very low, compared to other products such as bread substituted with turmeric flour [47], where the amount of this compound was related to the color of the final product and the antioxidant capacity of curcumin, which means that the yogurts are not a representative dietary source of this compound. 

The results obtained during the color and curcumin quantification in the yogurts showed the potential of the CLS suspension as a color additive for food products, in this particular case, in yogurt products. Further investigations may evaluate its application in other types of food matrices since the characteristic pH value of yogurts, could contribute to the stability of the color suspension [48].

### 3.5. Rheology

In yogurts, the obtained structure during milk fermentation determines the rheological properties of the final product, which can be affected by the addition of flavors and stabilizers (Walstra et al., 2006). Additionally, in yogurts, viscosity is considered an important descriptor for textural perception [49]. The apparent viscosity of the yogurts as a function of shear rate is shown in Figure 5. 

For all the samples, the apparent viscosity decreased with the increase in shear rate, indicating a shear-thinning fluid behavior. This could be attributed to the disentanglement of the chains under the shear field and the breaking of the structure [50]. Additionally, in the figure, two groups can be observed, one formed by the 0% CLS and 1% CLS samples and the other one by the 1.25% CLS and 1.5% CLS yogurts. Thus, larger concentrations of CLS are related to yogurts with higher resistance to flow after 24 h of storage. This is confirmed by the apparent viscosity in 50 s^−1^ ($\eta_{50}$) (Table 3), which has been reported as a representative oral shear rate. The obtained values are similar to those reported for stirred yogurts [25,51], that is, the addition of the CLS suspension in the yogurts did not negatively affect their viscosity. Samples consistency can be influenced by the interactions between the particles [52]. According to the results observed in yogurts with CLS, it is possible to establish that the components of the suspension interact with the components of the yogurt [50], as previously established during the syneresis analysis. As stated in the methodology, the upward curves were fitted to the Ostwald-de-Waele model. The resulting parameters are shown in Table 3.

According to the results, the model fitted the experimental values for all the yogurts with R^2^ values higher than 0.9. The flow index of the yogurts, showed a reduction with the increase in CLS, indicating a more viscous nature in the products, while the consistency index (K) increased, indicating greater firmness of the sample. This is attributed to the weakening of the interaction between the protein-peptide aggregates of yogurt, by adding the CLS suspension [25]. The previous behavior has been reported for yogurts added with apple pomace, soy, and quinoa [25,50,53]. The thixotropic phenomenon of fluids can be explained by the change in its viscosity when an external shear force is applied. In the beginning, the gel network is able to resist the shear force, and the breakdown occurs after a critical shear rate is applied. This behavior is estimated by the difference between the areas under the flow curves [34]. The area of the hysteresis loop (ΔA) for the samples is shown in Table 3. A statistically significant increase was seen. That is, an increment in the concentration of the CLS suspension generates yogurts more susceptible to structural breakdown when shear stress is applied, and after shearing, the restructuring into a network of the protein aggregates was more difficult [18,25]. Finally, it is important to remember that the results presented on the viscosity of the samples were determined after 24 h of their preparation, and as reported in the literature, an improvement in the consistency and viscosity of yogurt occurs in the first moments of storage, which is related to the solidification of the gel during cooling [54]. However, as storage progresses, yogurt may lose consistency and generate changes in rheological properties [27,55]. According to the aforementioned, in future investigations the effect of the addition of CLS on the viscosity during storage time could be evaluated. 

### 3.6. Sensory Analysis 

Initially, 50 panelists participated during the sensory analysis of yogurts. However, due to errors during the survey diligence, including the lack of personal information or incomplete responses, the results presented below in Table 4 correspond to 40 people (16 men and 24 women) aged between 19 and 33 years.

According to the results, the addition of the CLS did not affect the color and texture of the yogurts; while the 1.5% CLS sample showed a decrease in the evaluation of odor and taste, presenting notes “neither liked nor disliked”, which is still considered a satisfactory evaluation [56]. These results are associated with the sensory attributes of CL, which is characterized by presenting pungent and bitter flavors [57], associated with the presence of terpenoids [47]. Furthermore, the rhizome presents a characteristic odor, due to the presence of ar-turmerone [57]. This behavior was observed in other products with turmeric. For example, during the substitution of wheat flour for turmeric powder in bread making [47] and during incorporation of turmeric powder in Korean jellies or “Yanggaeng” [58]. It is important to establish that additives from vegetable sources tend to impart characteristic odors and flavors due to the presence of volatile compounds associated with the initial raw material [57]. In the case of the obtained yogurts, although there is a tendency to reduce the acceptance of the color and smell of the samples as the concentration of CLS increases, it is important to note that all the samples have the same general acceptance, associated with an acceptable level of satisfaction. As mentioned before, viscosity is an important descriptor of the texture in yogurt, and the viscosity values (η_50_) for all the samples were found in the range reported as acceptable for this type of yogurt, which is associated with the positive evaluation of the texture by the panelists. Additionally, according to the sensory results, the obtained values are similar to those reported for stirred yogurts [25,51]. That is, the addition of the CLS suspension in the yogurts did not negatively affect their viscosity.

According to the results obtained during this investigation, it is possible to establish that the CLS suspension can be used as a coloring additive in stirred yogurts since the yellow color remains stable up to 21 days of storage. Furthermore, the addition of 1 wt. % or 1.25 wt. % of CLS allows the development of a product that meets the physicochemical parameters established in the regulations (pH and titratable acidity) and has acceptable sensory evaluation; however, it is important to remark the appearance of syneresis in the last days of storage, behavior that has been reported with the addition of other ingredients in this type of food product. 

## 4. Conclusions

The development of additives for the food industry has different challenges. In this case, a CLS suspension was used as a color additive during a stirred yogurt formulation. The obtained results allow us to establish that the suspension does not affect quality factors such as pH and acidity; however, the increase in the CLSs concentration in yogurts was associated with an increase in syneresis, an undesired characteristic that is associated with the loss of the structure characteristic of this type of product. Additionally, the CLS suspension can impart a yellow color to yogurts, and this characteristic remains stable during storage. The previous result is directly associated with the stability of the curcumin observed in all the products since it is responsible for the yellow coloration of turmeric. Furthermore, according to the sensory characterization, the addition of 1% or 1.25% of CLS allows the development of a yogurt with adequate sensory quality. Finally, the observations achieved during this investigation show the potential of the raw turmeric rhizome to develop color agents for the food industry.

## Figures and Tables

**Figure 1 molecules-27-00946-f001:**
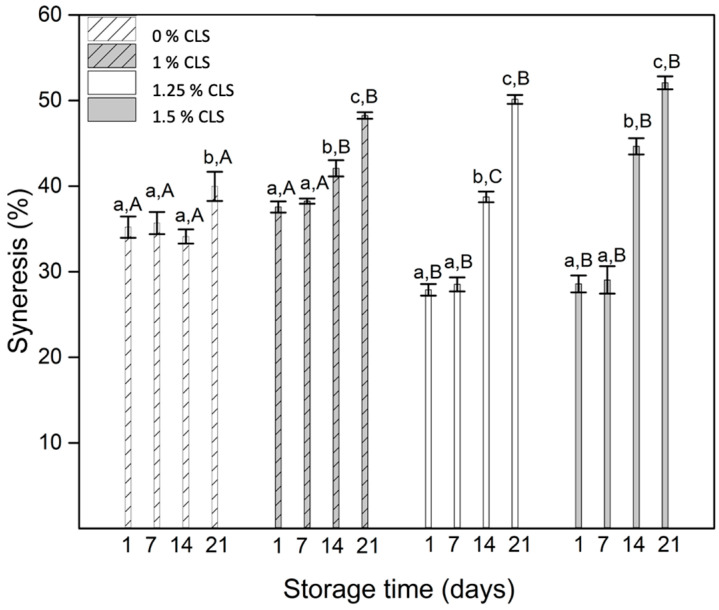
Syneresis of the yogurts with and without CLS. a–c for each sample indicates statistically significant difference (*p* < 0.05) during storage. A–C for each storage day indicates statistically significant difference (*p* < 0.05) between samples.

**Figure 2 molecules-27-00946-f002:**
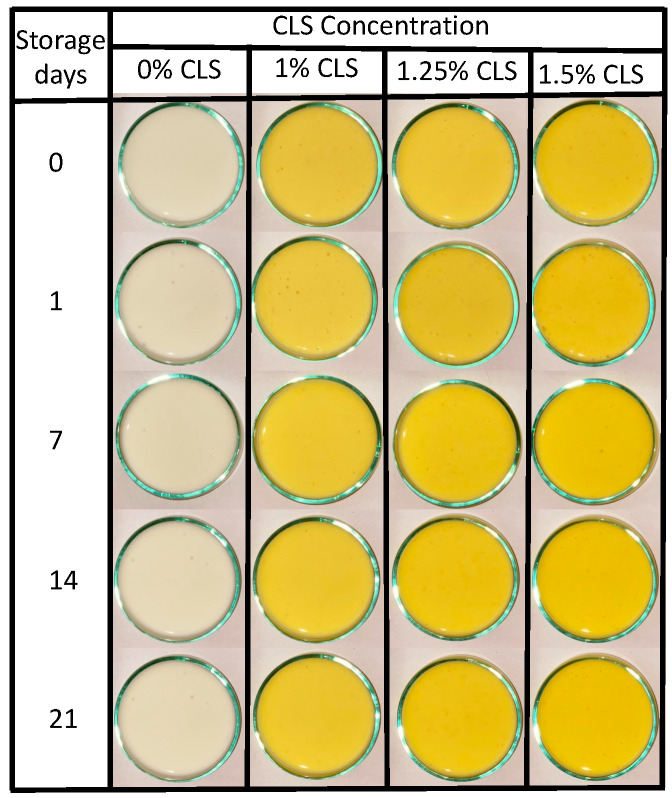
Pictures of the yogurts with and without CLS during storage time.

**Figure 3 molecules-27-00946-f003:**
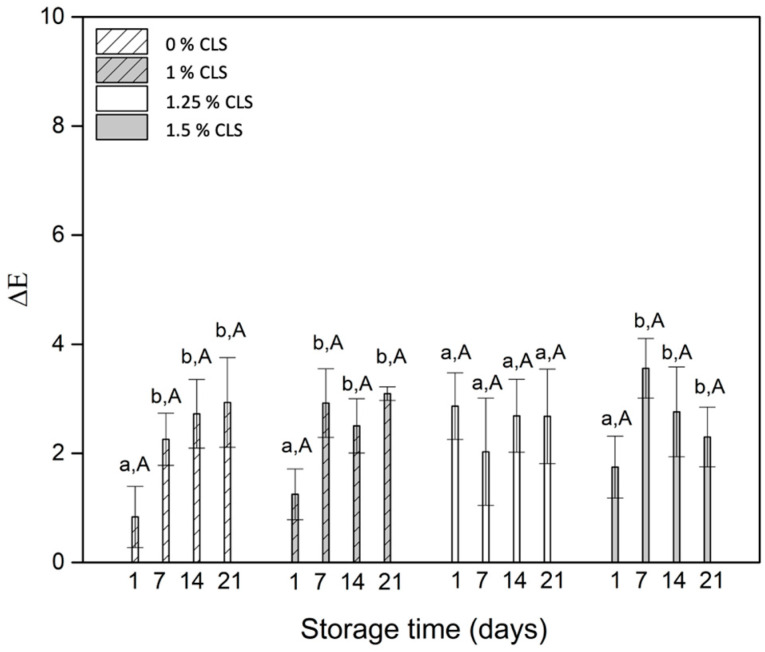
Total color change of the yogurts with and without CLS. a,b for each sample indicates statistically significant difference (*p* < 0.05) during storage. A for each storage day indicates no statistically significant difference (*p* < 0.05) between samples.

**Figure 4 molecules-27-00946-f004:**
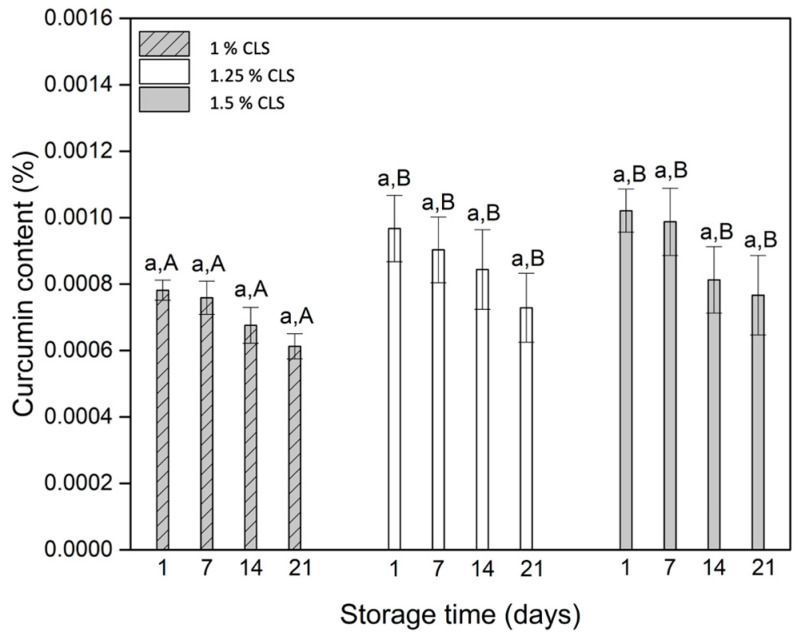
Curcumin content in the yogurts with and without CLS. a for each sample indicates no statistically significant difference (*p* < 0.05) during storage. A,B for each storage day indicates statistically significant difference (*p* < 0.05) between samples.

**Figure 5 molecules-27-00946-f005:**
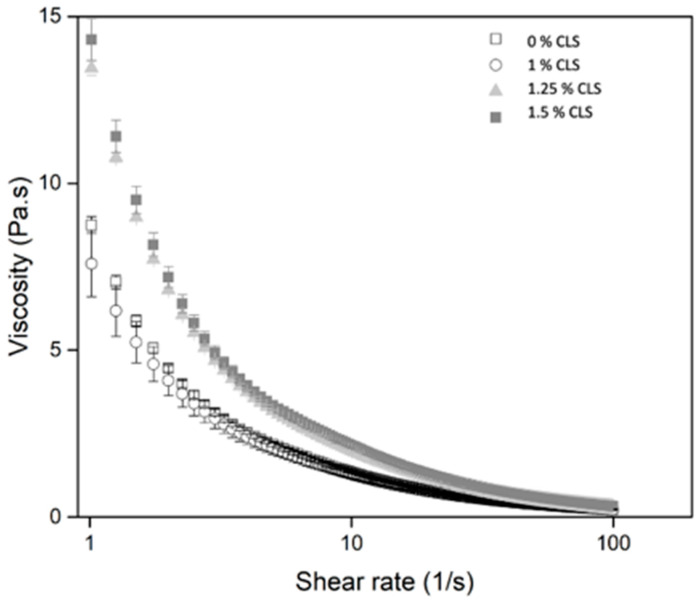
Apparent viscosity of the yogurts with and without CLS.

**Table 1 molecules-27-00946-t001:** pH and titratable acidity of the yogurts with and without CLS.

Sample	Storage Time (Days)	pH	Titratable Acidity (wt. % Actic Acid)
0% CLS	1	4.410 ± 0.017 **a,A**	0.938 ± 0.044 **a,A**
	7	4.307 ± 0.067 **a,A**	0.914 ± 0.012 **a,A**
	14	4.383 ± 0.015 **a,A**	0.919 ± 0.001 **a,A**
	21	4.363 ± 0.047 **a,A**	0.913 ± 0.005 **a,A**
1% CLS	1	4.347 ± 0.127 **a,A**	0.900 ± 0.003 **a,A**
	7	4.357 ± 0.025 **a,A**	0.931 ± 0.031 **a,A**
	14	4.340 ± 0.069 **a,A**	0.932 ± 0.010 **a,A**
	21	4.330 ± 0.020 **a,A**	0.912 ± 0.019 **a,A**
1.25% CLS	1	4.373 ± 0.015 **a,A**	0.936 ± 0.011 **a,A**
	7	4.367 ± 0.040 **a,A**	0.918 ± 0.004 **a,A**
	14	4.360 ± 0.061 **a,A**	0.919 ± 0.001 **a,A**
	21	4.353 ± 0.045 **a,A**	0.887 ± 0.074 **a,A**
1.5% CLS	1	4.340 ± 0.036 **a,A**	0.902 ± 0.022 **a,A**
	7	4.313 ± 0.015 **a,A**	0.906 ± 0.038 **a,A**
	14	4.327 ± 0.006 **a,A**	0.923 ± 0.002 **a,A**
	21	4.320 ± 0.026 **a,A**	0.948 ± 0.020 **a,A**

**a** for each simple indicates no statistically significant difference (*p* < 0.05) during storage. **A** for each storage day indicates no statistically significant difference (*p* < 0.05) between samples.

**Table 2 molecules-27-00946-t002:** Color of the yogurts with and without CLS.

Sample	Storage Time (Days)	L*	a*	b*
0% CLS	0	88.122 ± 1.600 **a,A**	−2.265 ± 0.306 **a,A**	6.590 ± 0.989 **a,A**
	1	88.705 ± 0.434 **a,A**	−2.242 ± 0.250 **a,A**	6.740 ± 0.490 **a,A**
	7	90.285 ± 1.139 **a,A**	−2.425 ± 0.208 **a,A**	6.295 ± 0.238 **a,A**
	14	90.687 ± 0.456 **a,A**	−2.120 ± 0.810 **a,A**	6.253 ± 0.212 **a,A**
	21	90.885 ± 0.725 **a,A**	−2.105 ± 0.120 **a,A**	6.178 ± 0.760 **b,A**
1% CLS	0	85.565 ± 1.623 **a,A**	−3.203 ± 0.244 **a,A**	37.905 ± 0.299 **a,B**
	1	85.618 ± 0.506 **a,A**	−3.700 ± 0.306 **a,B**	38.648 ± 0.721 **a,B**
	7	86.288 ± 2.029 **a,A**	−4.225 ± 0.316 **a,B**	38.355 ± 1.726 **a,B**
	14	87.530 ± 0.741 **a,A,B**	−4.433 ± 0.453 **a,B**	37.873 ± 0.508 **a,B**
	21	87.890 ± 0.439 **a,B**	−4.518 ± 0.590 **a,B**	36.585 ± 0.392 **a,B**
1.25% CLS	0	84.327 ± 0.649 **a,A**	−2.723 ± 0.491 **a,A**	41.893 ± 0.279 **a,C**
	1	85.225 ± 2.438 **a,A**	−2.888 ± 0.374 **a,A,B**	42.745 ± 1.283 **a,C**
	7	85.897 ± 0.820 **a,A**	−3.067 ± 0.287 **a,A**	43.015 ± 0.344 **a,B,C**
	14	86.041 ± 1.858 **a,B**	−3.710 ± 0.215 **a,A,B**	42.976 ± 0.184 **a,C**
	21	86.109 ± 0.858 **a,B**	−3.875 ± 0.143 **a,B**	43.396 ± 0.184 **a,C**
1.5% CLS	0	85.313 ± 1.252 **a,A**	−2.298 ± 0.433 **a,A**	44.885 ± 0.558 **a,D**
	1	85.675± 1.572 **a,A**	−2.310 ± 0.267 **a,A**	45.075 ± 0.715 **a,C**
	7	85.078 ± 1.599 **a,A**	−3.078 ± 0.162 **b,A**	45.805 ± 3.039 **a,C**
	14	86.945 ± 0.620 **a,A,B**	−3.160 ± 0.516 **b,A,B**	46.019 ± 1.775 **a,C**
	21	87.013 ± 0.335 **a,B**	−3.410 ± 0.051 **b,B**	45.914 ± 0.368 **a,D**

**a**,**b** for each sample indicates statistically significant difference (*p* < 0.05) during storage. **A**–**D** for each storage day indicates statistically significant difference (*p* < 0.05) between samples.

**Table 3 molecules-27-00946-t003:** Rheological parameters of yogurts with and without CLS.

CLS	$\eta_{50}$	K (Pa.s^n^)	n	R^2^	ΔA (Pa.s)
0%	0.377 ± 0.024 **A**	6.186 ± 0.263 **A**	0.716 ± 0.006 **A**	0.996	411.562 ± 53.856 **A**
1%	0.386 ± 0.011 **B**	7.663 ± 0.270 **B**	0.766 ± 0.003 **B**	0.995	407.714 ± 21.919 **A**
1.25%	0.573 ± 0.009 **C**	11.792 ± 0.079 **C**	0.777 ± 0.003 **C**	0.995	727.393 ± 21.267 **B**
1.50%	0.609 ± 0.025 **D**	12.688 ± 0.491 **D**	0.781 ± 0.001 **D**	0.995	785.999 ± 30.875 **C**

**A**–**D** indicates statistically significant difference (*p* < 0.05) between samples.

**Table 4 molecules-27-00946-t004:** Sensory scores of the yogurts with and without CLS (n = 40).

CLS	Color	Odor	Texture	Flavor	Overall Acceptability
1%	7.075 ± 1.248 **a**	6.650 ± 1.292 **a**	6.800 ± 1.713 **a**	6.150 ± 2.007 **a**	6.425 ± 1.517 **a**
1.25%	6.850 ± 1.545 **a**	6.375 ± 1.690 **a**	6.575 ± 1.752 **a**	6.275 ± 1.867 **a**	6.350 ± 1.642 **a**
1.50%	7.225 ± 1.250 **a**	4.925 ± 1.269 **b**	6.925 ± 1.700 **a**	5.050 ± 2.459 **b**	5.850 ± 1.777 **a**

**a**,**b** for each column indicates statistically significant difference (*p* < 0.05).

## Data Availability

Not applicable.

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
