# Peer review of "The Effects of Adding a Gel-Alike Curcuma longa L. Suspension as Color Agent on Some Quality and Sensory Properties of Yogurt"

_molecules, 2022, doi:10.3390/molecules27030946_

Round 1

Reviewer 1 Report

The manuscript “The effects of adding a gel-alike Curcuma longa L. suspension as color agent on some quality and sensory properties of yogurt” present the results of the evaluation of the possibility and effects of application of the entire rhizome of Curcuma longa (CL) as a yoghurt colourant. This is a well-written, innovative and interesting manuscript with application potential.  I have only some comments (see below) that could be taken into consideration before publication:
-    Line 32-33: in this sentence, instead of ‘… compared to synthetic dyes ’ it would be better to write ‘The application of natural additives during food formulations has attracted considerable interest because of their safety compared to synthetic ones.’
-    Line 38 – please explain the abbreviation ‘wt’ when it appears in the text for the first time
-    Line 110: Add information on the curcumin standard producer
-    Line 135: ‘… had to sign an informed consent’ - please explain whether the panellist consent was voluntary

Author Response

  • Line 32-33: in this sentence, instead of ‘… compared to synthetic dyes ’ it would be better to write ‘The application of natural additives during food formulations has attracted considerable interest because of their safety compared to synthetic ones. A/: The introduction was re-written according to the academic editor suggestion.
  • Line 38 – please explain the abbreviation ‘wt’ when it appears in the text for the first time. A/: According to the international nomenclature, “wt” means weight percent.
  • Line 110: Add information on the curcumin standard producer. A/:  Information about the standard is added in the manuscript as suggested by the reviewer (Line 110).
  • Line 135: ‘… had to sign an informed consent’ - please explain whether the panellist consent was voluntary. A/: The suggestion by the reviewer is made in the manuscript (Lines 138 to 139).

Reviewer 2 Report

Manuscript number: molecules-1512378-peer-review-v1

Article Type: Article

Title: The effects of adding a gel-alike Curcuma longa L. suspension as color agent on some quality and sensory properties of yogurt

Comments:

Line 118: please check the unit for apparent viscosity; are You sure that is correct? The same situation with parameter K (Line 119)

Line 118: why do You use g for shear rate and after that use gamma in the equation below? Should be letter gamma but with dot; in the eq. should be tau  not eta

Table 3. Please put the units in the table; K should have Pa·sn, not Pa·s.

Table 3. ΔA – what is? It is not explained in the method and in Table 3. What is the unit? Is it value of area between flow curves?

Line 325. Consistency index here is small k, but in the Table 3 is capital letter. Should be in he text the same, small or capital, not both.

Table 3. Please explain why table 3 shows negative values of  flow index. Do You describe downward flow curve? The value of the flow index should be from 0 to 1.

Recommendation: REJECT

In my opinion too many errors appear in the manuscript, for example in rheological studies.

Author Response

  • Line 118: please check the unit for apparent viscosity; are You sure that is correct? The same situation with parameter K (Line 119). A/: The units for apparent viscosity and consistency index are completed in the materials section according to the reviewers comments (Lines 121 and 122).
  • Line 118: why do You use g for shear rate and after that use gamma in the equation below? Should be letter gamma but with dot. A/: Parameters in the equation and in the text are changed, as required by the reviewer (Lines 121 and 127).
  • Table 3. Please put the units in the table; K should have Pa·sn, not Pa·s. A/: The suggestion made by the reviewer is made in the text (Table 3).
  • Table 3.Δ A – what is? It is not explained in the method and inTable 3. What is the unit? Is it value of area between flow curves?. A/: Information about ΔA is added in the materials and methods section (Lines 120-121) as suggested by the reviewer.
  • Line 325. Consistency index here is small k, but in the Table 3 is capital letter. Should be in he text the same, small or capital, not both. A/: The “K” from consistency index is changed in the manuscript as suggested by the reviewer (Line 340).  
  • Table 3. Please explain why table 3 shows negative values of  flow index. Do You describe downward flow curve? The value of the flow index should be from 0 to 1. A/: The comment of the reviewer is completely valid.  The values reported for the flow index are for 0 to 1 and they were changed (Table 3). This was a typo error.

Reviewer 3 Report

1. The content of introduction should be extended. The more related articles should be cited and introduced in this section.

2. The method about data analysis should be added in the end of Materials and Method section.

3. The important data of results should be introduced in the results discussion section.

4. The effect mechanism of suspensions on the quality of yogurt should be discussed deeply by cited more related articles.

5. The conclusion is too long and should be shorted. 

Author Response

  • The content of introduction should be extended. The more related articles should be cited and introduced in this section. A/: Introduction section was re-written according to the reviewer and the academic editor suggestions.
  • The method about data analysis should be added in the end of Materials and Method section. A/: Information about the statistical analysis is added in the manuscript as suggested by the reviewer(Lines 143 to 148).
  • The important data of results should be introduced in the results discussion section. A/: In the present manuscript, we did not separate the discussion section from the results. All the information is presented in one section. This form of presentation is valid for the editorial board.
  • The effect mechanism of suspensions on the quality of yogurt should be discussed deeply by cited more related articles. A/: New information about the stability mechanism is added in the results section, as suggested by the reviewer (Lines 211 to 217). Additionally, bibliographic data was added to support the discussion.
  • The conclusion is too long and should be shorted. A/:  Conclusions section is changed, according to the reviewer comments (Lines 392).

Round 2

Reviewer 2 Report

I accept the manuscript in present form.

Reviewer 3 Report

The presentaiotn of result data should be improved in the language according to the result figure, which was presented too simple.